# Humoral Response in Hemodialysis Patients Following COVID-19 Vaccination and Breakthrough Infections during Delta and Omicron Variant Predominance

**DOI:** 10.3390/vaccines10040498

**Published:** 2022-03-24

**Authors:** Rajkumar Chinnadurai, Henry H. L. Wu, Eleanor Cox, Jayne Moore, Toni Clough, Elizabeth Lamerton, Rosie Donne, Edmond O’Riordan, Dimitrios Poulikakos

**Affiliations:** 1Department of Renal Medicine, Northern Care Alliance NHS Foundation Trust, Salford M6 8HD, UK; eleanor.cox@nca.nhs.uk (E.C.); jayne.moore@nca.nhs.uk (J.M.); toni.clough@nca.nhs.uk (T.C.); elizabeth.lamerton@nca.nhs.uk (E.L.); rosie.donne@nca.nhs.uk (R.D.); edmond.oriordan@nca.nhs.uk (E.O.); dimitrios.poulikakos@nca.nhs.uk (D.P.); 2Faculty of Biology, Medicine and Health, University of Manchester, Manchester M13 9PL, UK; henrywu96@yahoo.com; 3Department of Renal Medicine, Lancashire Teaching Hospitals NHS Foundation Trust, Preston PR2 9HT, UK

**Keywords:** COVID-19 vaccination, humoral response, hemodialysis, breakthrough infections, Delta variant, Omicron variant

## Abstract

Background: The advancement of COVID-19 vaccination programs globally has been viewed as an integral strategy to reduce both the number of COVID-19 cases and consequential complications of COVID-19, particularly for high-risk patient groups. There are limited data on the antibody response and protection from disease infection and severity in patients requiring hemodialysis (HD) following COVID-19 vaccination during the Delta and Omicron variant predominance. We conducted a study aiming to evaluate humoral immunity derived from two different COVID-19 vaccines administered to our in-centre HD population and investigated the characteristics of breakthrough COVID-19 infections occurring post-vaccination within this population. Methods: This is a prospective observational study including patients receiving HD at Salford Royal Hospital. The first and second doses of COVID-19 vaccinations (Pfizer BioNTech BNT162b2 or Oxford AstraZeneca ChAdOx1 nCoV-19) were administered to this patient cohort since January 2021. The incidence of any breakthrough COVID-19 infections occurring in double vaccinated patients between 1 April 2021 and 15 January 2022 was recorded. Patients were screened weekly with nasal and pharyngeal nasopharyngeal swabs for real-time Reverse Transcription Polymerase Chain Reaction (rRT-PCR) for COVID-19, whilst SARS-CoV-2 antibody testing was performed alongside monthly routine HD bloods. Results: Four hundred eleven patients receiving HD were included in this study, of which 170 of 178 patients (95.5%) with available data on antibody status following two doses of the Pfizer BioNTech BNT162b2 vaccination had detectable antibody response, whilst this was the case for 97 of 101 patients (96.1%) who received two doses of the Oxford AstraZeneca ChAdOx1 nCoV-19 vaccine. For 12 seronegative patients who received a booster vaccine (third dose), nine seroconverted, while one remained negative and two were not tested. No statistically significant differences were observed with regards to antibody status between those receiving Pfizer BioNTech BNT162b2 and Oxford AstraZeneca ChAdOx1 nCoV-19 vaccines. Sixty-three of 353 patients with two doses of COVID-19 vaccination had breakthrough COVID-19 infection (40 during Delta and 23 during Omicron variant predominance). Of the 40 patients during the delta period, five were admitted into hospital and there were two reported deaths due to COVID-19-related illness. There were no COVID-19 associated hospitalizations or deaths during the Omicron variant predominance. Conclusions: The vast majority of HD patients who received two doses of the Pfizer BioNTech BNT162b2 or Oxford AstraZeneca ChAdOx1 nCoV-19 vaccinations developed detectable antibody responses. Our results support the value of booster vaccination with mRNA-based COVID-19 vaccine in HD patients and highlight the need for ongoing surveillance programmes with rRT-PCR and antibody testing for timely detection of positive cases.

## 1. Introduction

Patients receiving in-centre hemodialysis (HD) are more vulnerable to COVID-19 infection, with a high risk of transmission within the dialysis environment [1]. Such patients will need to attend for life-sustaining treatment mostly thrice weekly, often requiring use of shared or public transport to travel to dialysis units. Furthermore, patients receiving HD usually have multiple co-morbidities linked to poor COVID-19 outcomes. A significant proportion of this patient group has a history of long-term immunosuppression prior to dialysis, which further increases their risks of hospitalization and mortality following COVID-19 infection [2].

The United Kingdom (U.K.) Kidney Association defined patients receiving HD as clinically extremely vulnerable from COVID-19 infection and recommended these patients receive COVID-19 vaccinations as soon as possible, irrespective of their age [3]. Studies have reported comparable or lower humoral response to messenger ribonucleic acid (mRNA) vaccine in HD patients compared to healthy subjects, and one U.K. study evaluating neutralizing antibodies showed suboptimal response to the Oxford AstraZeneca ChAdOx1 nCoV-19 vaccine [4,5,6,7].

The clinical aim of determining immune responses to COVID-19 vaccination is to provide correlation with protection against COVID-19 infection and disease severity in these patients and guide effective vaccination strategies. To date, there are sparse data regarding breakthrough infections from Delta and the recently emerged Omicron variant following COVID-19 vaccination in HD patients [8].

To this end, we conducted a study with two major aims: (1) To evaluate humoral immunity derived from two different COVID-19 vaccines (Pfizer BioNTech BNT162b2 and Oxford AstraZeneca ChAdOx1 nCoV-19) administered to the in-centre HD population and (2) to investigate the characteristics of breakthrough COVID-19 infections occurring post-vaccination within this patient population identified through our surveillance programme.

## 2. Materials and Methods

### 2.1. Study Population

This is a prospective observational cohort study for patients receiving in-centre HD in the main and satellite dialysis units of Salford Royal Hospital, a tertiary nephrology centre serving a population of 1.55 million in North-West UK. The summary of patient recruitment into this study is illustrated in Figure 1. Data including demographic information, co-morbidities, immunosuppression status, COVID-19 vaccination details, and COVID-19 antibody status pre- and post-vaccination were collated from our local electronic patient records and analyzed. Information on each patient’s COVID-19 antibody status was gathered until 30 November 2021.

The incidence of any breakthrough COVID-19 infections occurring in double vaccinated patients between 1 April 2021 and 15 January 2022 (end of follow up) was recorded, with positive COVID-19 infection status defined by positive real time reverse transcription polymerase chain reaction (rRT-PCR) in surveillance testing. COVID-19-related death was defined as death due to COVID-19 pneumonitis reported on the death certificate. COVID-19-related hospitalization was defined as hospitalization due to COVID-19 and was adjudicated by two clinicians (D.P., R.C.).

### 2.2. Assays

All HD patients included in this study had been receiving monthly COVID-19 immunoglobulin G (IgG) antibody surveillance testing with Siemens’ COV2T immunoassay which is used to detect total antibodies (IgG and IgM) to SARS-CoV-2. This is a qualitative assay with an index ≥1.0 being considered as positive based on the manufacturer-determined cut-off. The sensitivity and specificity for the assay were reported at 98.1% and 99.9%, respectively [9]. All patients also had a weekly surveillance COVID-19 rRT-PCR throat and nasal swabs since August 2020 [9].

### 2.3. Vaccination

The first and second doses of COVID-19 vaccinations (Pfizer BioNTech BNT162b2 or Oxford AstraZeneca ChAdOx1 nCoV-19) were administered to this patient cohort from January 2021. The vaccinations were offered either in the community or during routine dialysis sessions coordinated by the renal department. The third dose of vaccine administered from September 2021 was Pfizer BioNTech BNT162b2 vaccine, irrespective of the primary vaccination type.

### 2.4. Statistical Analysis

Simple descriptive statistics were used to summarize demographic and clinical data, and data relating to COVID-19 vaccination and antibody status. We evaluated the distribution of characteristics based on antibody status for HD patients who have had two doses of their COVID-19 vaccinations by June 2021. Delta proxy period was defined as 1 April 2021 to 20 December 2021 and start date of Omicron proxy period was determined on 21 December 2021, when the Omicron strain became dominant in the region [10]. Breakthrough COVID-19 infection in patients who had the double dose of the primary vaccination was examined separately during the Delta proxy and Omicron proxy period. Throughout the analysis, categorical variables were expressed as ‘number (%)’ and its *p*-value by Chi-Square test. Continuous variables were expressed as ‘median (interquartile range)’ and *p*-value by the Mann–Whitney U test. Binary logistic regression analysis through a multi-variate statistical model was used to evaluate for associations between variables and attenuated antibody response following two doses of COVID-19 vaccination. All statistical analyses were performed on SPSS version 24.0 (registered to the University of Manchester).

This study was registered with the Northern Care Alliance Research and Innovation department (Ref. No.: S21HIP08). As this is an observational study with complete anonymization of patient identification details, there was no indication to consent for each individual patient in this study.

## 3. Results

### 3.1. Baseline Characteristics

Amongst 411 HD patients included in this study, the median age was 61 years (49–72) (Table 1). The majority of patients were of Caucasian ethnicity (65.9%) and male (64%). Co-morbidities with significant prevalence included hypertension (71.8%), diabetes mellitus (49.1%), and cardiovascular disease (28.7%). In this cohort, 15.3% of patients were previous kidney transplant recipients, and around 18% were receiving immunosuppressive medications.

### 3.2. Vaccination Status

The first Pfizer BioNTech BNT162b2 and Oxford AstraZeneca ChAdOx1 nCoV-19 vaccinations were administered to this patient cohort from January 2021, and the majority of patients included in this study received two doses of their COVID-19 vaccinations by 30 June 2021. 27% of the cohort had positive COVID-19 IgG antibodies prior to vaccination. Three hundred seventy-seven (91.7%) and 353 (85.8%) HD patients, respectively, received their first and second doses of COVID-19 vaccination by June 2021 (Figure 2A). Of the 377 HD patients who received at least one dose of COVID-19 vaccination by this time, 236 patients (62.6%) received the Pfizer BioNTech BNT162b2 vaccination, 135 patients (35.8%) received the Oxford AstraZeneca ChAdOx1 nCoV-19 vaccine, and details of the brand of vaccine for 6 patients (1.59%) was not traceable.

### 3.3. Antibody Status

Of the 353 HD patients who received two doses of COVID-19 vaccination by June 2021, 269 of the 281 patients (96.3%) with available data on antibody status following vaccination developed antibodies to SARS-CoV-2 by the end of July 2021 (Figure 2B). By July 2021, 170 of 178 patients (95.5%) with available data on antibody status following both doses of the Pfizer BioNTech BNT162b2 vaccination had antibody response, whilst this was the case for 97 of 101 patients (96.1%) who received both doses of the Oxford AstraZeneca ChAdOx1 nCoV-19 vaccine. Twelve HD patients did not develop any antibodies despite receiving two vaccination doses (Table 2A). Reviewing the characteristics within this sub-group compared to those who developed antibodies, patients who did not develop immunity were older (69.5 vs. 62 years, *p* = 0.09). No further statistically significant differences were observed between the two groups. Logistic regression analysis through a multi-variate statistical model did not identify any independent significant associations between variables and lack of antibody response following two doses of COVID-19 vaccination (Table 2B). Further to the third booster dose of vaccine, 9 of the 12 developed antibodies to SARS-CoV-2, while one remained negative and two were not tested.

### 3.4. Breakthrough Infections

Amongst the 353 patients who received two doses of COVID-19 vaccination by June 2021, 63 had breakthrough COVID-19 infection identified by surveillance PCR following their second vaccination dose. The comparison of baseline characteristics and type of primary vaccination between patients with and without breakthrough COVID-19 infection did not reveal any significant differences between the two groups. The pre-vaccination antibody status was negative in a greater proportion of patients who had breakthrough infections (*p* = 0.005) (Table 3). The number of breakthrough infections increased with greater time from primary vaccination during the Delta proxy period (Figure 3).

A comparison of the patient characteristics, including vaccination and antibody status, in relation to breakthrough infections during the Delta proxy (1 April 2021 to 20 December 2021) and Omicron proxy period (21 December 2021 to 15 January 2022) found no significant differences between the groups. There were five (12.5%) hospitalizations and two (5%) deaths reported in the Delta proxy period but no death or hospitalization during the Omicron proxy period, and most patients (74%) were triple vaccinated (Table 4).

## 4. Discussion

This study shows that the vast majority of HD patients generate detectable humoral response following COVID-19 vaccination with either Pfizer BioNTech BNT162b2 or Oxford AstraZeneca ChAdOx1 nCoV-19 vaccines. Most patients who did not develop detectable antibodies following two doses of primary vaccination seroconverted following the third dose. This cohort, compared to a recent study in the general population, exhibited similar hospitalization rates but higher mortality rates due to breakthrough COVID-19 infection during the Delta proxy period (12.5% vs. 12.7% and 5.0% vs. 1.1%, respectively) [11]. Infection from Omicron appears less severe in vaccinated HD patients. The major strength of the study is the accuracy of breakthrough infection detection based on routine surveillance weekly nasal and pharyngeal rRT-PCR.

Patients receiving HD historically have attenuated antibody responses following immunization compared to the general population, as reflected from data in hepatitis B and influenza vaccination studies [12,13,14]. However, mRNA-based COVID-19 vaccines appear to elicit a detectable antibody response in a substantially higher proportion of patients compared to hepatitis B or influenza vaccinations. Previously reported suboptimal levels of neutralizing antibodies following Oxford AstraZeneca ChAdOx1 nCoV-19 vaccine in this cohort pose the question of the relative effectiveness of this vaccine in HD patients [15]. Our study is the first to date to directly explore and compare antibody responses and breakthrough infections between an mRNA-based vaccine (Pfizer BioNTech BNT162b2) and a replication-defective viral-vectored vaccine (Oxford AstraZeneca ChAdOx1 nCoV-19) [7]. Although we did not measure neutralizing antibody levels, our results here suggest that there were no significant differences in detectable antibody response or breakthrough infections between the Pfizer BioNTech BNT162b2 and Oxford AstraZeneca ChAdOx1 nCoV-19 vaccines. In addition, the third dose of Pfizer BioNTech BNT162b2 resulted in seroconversion in patients who did not generate detectable antibody response following primary vaccination.

Another key finding from our study is that COVID-19 hospitalization rate during the Delta proxy period is shown to be 12.5%, which is similar to the recently reported hospitalization rate following BNT162b2 vaccination in the vaccinated general population [11]. The mortality rate has remained three times higher amongst the in-centre HD population compared to the general population during the Delta variant predominance period. There is a marked decline in the mortality rate for the in-centre HD population, from what was 10.7% to our report of 5% just prior to the introduction of the vaccination programme [9,16]. Furthermore, the hospitalization rate in our study was derived from asymptomatic weekly surveillance rRT-PCR screening, thus the hospitalization rate within symptomatic individuals is likely to have been higher and consequently, the rate is not directly comparable with the hospitalization rate in the study by Wang et al. [11] that included predominantly symptomatic cases. Our results also suggest that the incidence of COVID-19 breakthrough infections increases with passing time following primary vaccination and is in keeping with a recent nested case-control study by Anand et al. [14], which identified antibodies among patients on dialysis waning over time irrespective of prior SARS-CoV-2 infection status. The lack of COVID-19 hospitalizations or deaths during the Omicron period is encouraging and may be related to the high proportion of patients who had recently received the third dose that has been shown to increase the neutralization activity against Omicron and the milder natural course of the Omicron variant compared to Delta [17,18].

Our study did not indicate any specific risk factors, which independently predict attenuated antibody responses within the HD patient cohort following COVID-19 vaccination. Multiple recent studies highlighted the significant role of an individual’s immunosuppression in determining their vaccination-associated immune response [19,20,21]. In our study, the presence of immunosuppression was not statistically associated with detectable antibody response. We have not, however, used a quantitative antibody assay or neutralizing antibody measurement and were not able to examine the possible quantitative association of antibody response in relation to immunosuppression.

The findings from our study should be interpreted with some caution, due to a number of limitations. First, this was a single-centre study with small sample numbers in multiple sub-groups and imbalance between the number of patients receiving Pfizer BioNTech BNT162b2 and Oxford AstraZeneca ChAdOx1 nCoV-19 vaccinations. Our study depended only on the COVID-19 IgG antibody surveillance testing as a single measure of antibody response with the absence of quantitative data, measurement of neutralizing antibodies and markers of T-cell response. Furthermore, we did not have available genotype analysis of the COVID-19 positive swabs and relied on regional and national genotyping data to define Delta proxy and Omicron proxy periods. Despite these limitations, our cohort is very well characterized with prospective routine rRT-PCR testing ensuring breakthrough infections have been detected accurately even in asymptomatic patients.

## 5. Conclusions

Our study findings indicate that the vast majority of HD patients who received either the Pfizer BioNTech BNT162b2 or Oxford AstraZeneca ChAdOx1 nCoV-19 vaccinations developed detectable antibody responses, and the third dose of Pfizer BioNTech BNT162b2 led to seroconversion in patients with negative antibodies and mild infection during the Omicron proxy period. Our results support the value of booster vaccination with mRNA COVID-19 vaccine in HD patients and highlight the need for ongoing surveillance programmes with rRT-PCR and antibody testing for early detection and isolation of positive cases and for ongoing investigation of COVID-19 disease and response to vaccination in this vulnerable population. More studies are needed to derive further reliable biochemical markers to measure antibody response following COVID-19 vaccination.

## Figures and Tables

**Figure 1 vaccines-10-00498-f001:**
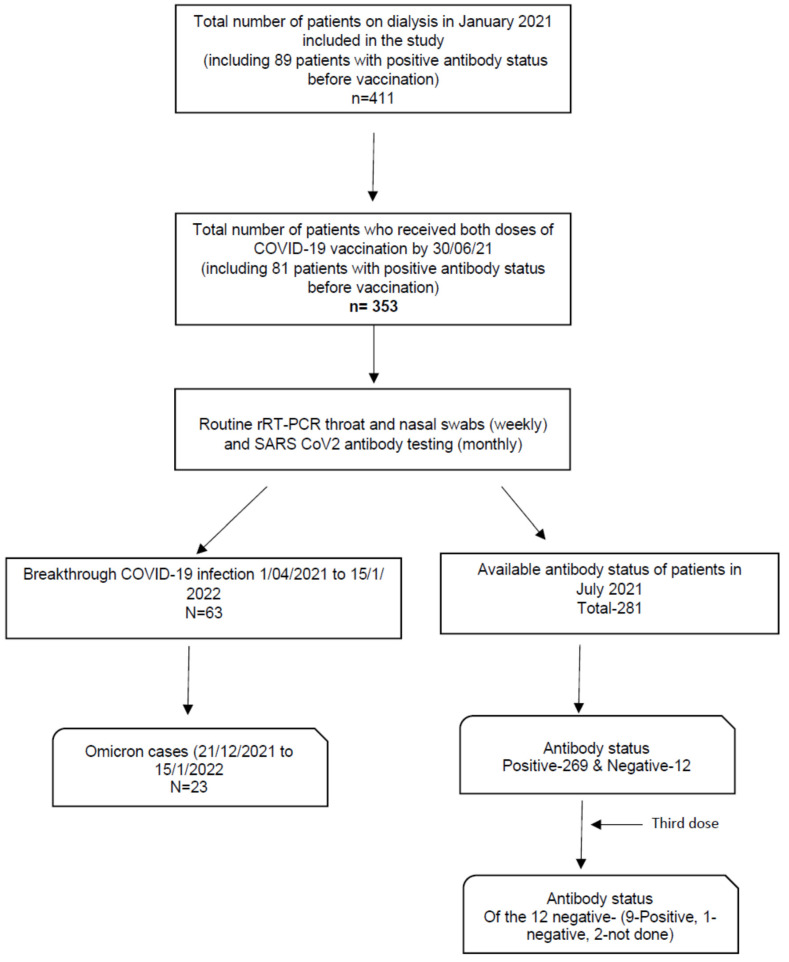
Flowchart for patient recruitment to the study.

**Figure 2 vaccines-10-00498-f002:**
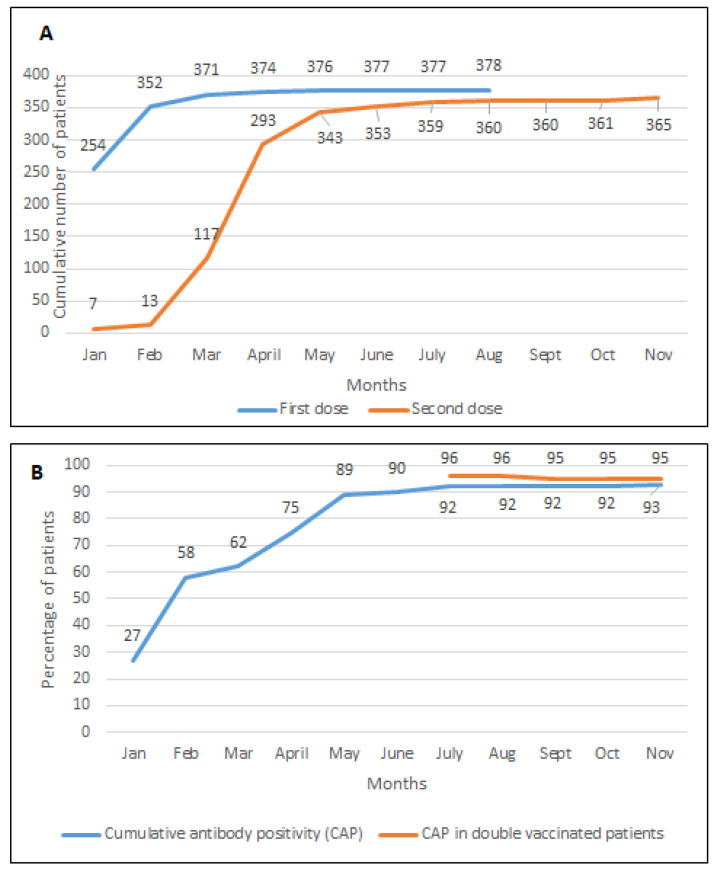
Monthly antibody status of the cohorts and timing of vaccinations. (**A**) Timing of vaccinations. (**B**) Cumulative antibody positivity in patients with available antibody status.

**Figure 3 vaccines-10-00498-f003:**
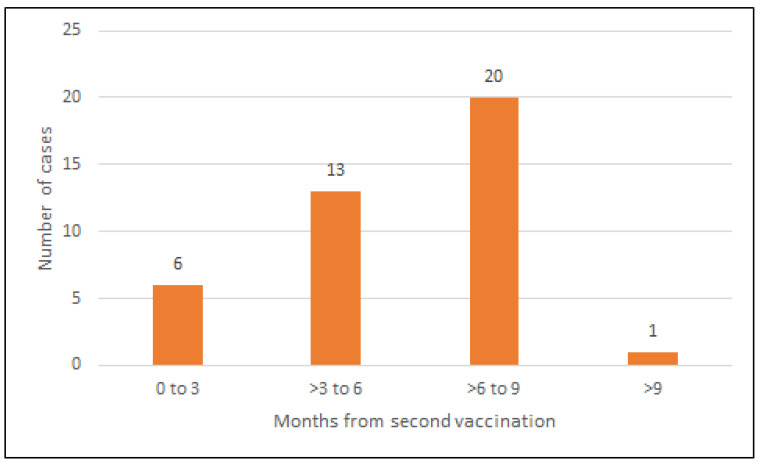
Time in months between the second dose of COVID-19 vaccination and the forty COVID-19 break-through infections before the Omicron proxy period (21 December 2021).

**Table 1 vaccines-10-00498-t001:** Baseline characteristics of the sample population in January 2021.

Total Patients	*n* = 411
Age, years	61 (49–72)
Ethnicity	
Caucasian	271 (65.9%)
Black Asian Minority Ethnic	140 (34.1%)
Gender, Male	263 (64%)
Diabetes Mellitus	202 (49.1%)
Hypertension	295 (71.8%)
Cardiovascular disease	118 (28.7%)
Cancer	39 (9.5%)
Previous kidney transplant	63 (15.3%)
Immunosuppression	
Current	34 (8.3%)
Previous	41 (10%)
None	336 (81.8%)
Vaccination status by June 2021	
Received first vaccination	377 (91.7%)
Received two vaccinations	353 (85.8%)
Refused vaccination/unknown	34 (8.3%)
Vaccination type (of the 377 vaccinated)	
Pfizer BioNTech	236 (62.6%)
Oxford AstraZeneca	135 (35.8%)
Unknown	6 (1.59%)

**Table 2 vaccines-10-00498-t002:** (**A**) Comparison of characteristics of patients who have received double dose vaccination by June 2021, based on the available antibody status in July 2021. (**B**) Binary logistic regression analysis (multi-variate model) to identify risk factors for attenuated antibody response following the second dose of vaccination.

Analysis A (Comparative Analysis)				Analysis B (Logistic Regression Analysis)	
Variables Total *n* = 281	Positive *n* = 269	Negative *n* = 12	*p*-Value	OR (95% CI)	*p*-Value
Age	62 (51–74)	69.5 (64–79)	0.093	0.97 (0.93–1.01)	0.18
Gender (Male)	173 (64.3%)	10 (83.3%)	0.176	0.36 (0.07–1.67)	0.19
Ethnicity (Caucasian)	173 (64.3%)	10 (83.3%)	0.200	2.62 (0.56–12.3)	0.21
Diabetes	127 (47.2%)	5 (41.6%)	0.706	1.25 (0.38–4.04)	0.71
Hypertension	209 (77.7%)	7 (58.3%)	0.120	2.48 (0.76–8.12)	0.13
Cardiovascular disease	78 (29%)	3 (25%)	0.765	1.22 (0.32–4.64)	0.76
Cancer	27 (10%)	3 (25%)	0.101	0.35 (0.08–1.31)	0.12
Immunosuppression	52 (19.3%)	3 (25%)	0.427	0.71 (0.18–2.74)	0.63
Previous transplant	44 (16.4%)	1 (8%)	0.458	2.15 (0.27–17.1)	0.46
Vaccination type			0.935		
Pfizer BioNTech	170 (63%)	8 (66.6%)			
Oxford AstraZeneca	97 (36%)	4 (33.3%)			
Unknown	2 (0.8%)	0			

Categorical variables expressed as number (%) and *p*-value by Chi-square test. Continuous variables expressed as median (interquartile range) and p-value by Mann–Whitney U test.

**Table 3 vaccines-10-00498-t003:** Comparison of characteristics of patients who have received double dose vaccination by June 2021, based on breakthrough COVID-19 infection between April 2021 and January 2022.

Variables Total *n* = 353	COVID Positive *n* = 63	COVID Negative *n* = 290	*p*-Value
Age	61 (49.5–73.5)	62 (52–73)	0.738
Gender (Male)	39 (61.9%)	200 (68.9%)	0.277
Ethnicity (Caucasian)	38 (60.3%)	188 (64.8%)	0.499
Diabetes	34 (53.9%)	137 (47.2%)	0.333
Hypertension	48 (76.1%)	210 (72.4%)	0.540
Cardiovascular disease	19 (30.1%)	85 (29.3%)	0.893
Cancer	5 (7.9%)	32 (11%)	0.467
Immunosuppression	10 (15.8%)	58 (20%)	0.452
Previous transplant	8 (12.6%)	45 (15.5%)	0.570
Pre-vaccination antibody status			
Positive	6 (9.5%)	75 (29%)	0.005
Negative	57 (90.5%)	215 (71%)	-
Antibody Status (July 2021)			
Positive	53 (84.1%)	216 (74.5%)	-
Negative	2 (3.1%)	10 (3.4%)	-
Unknown	8 (12.6%)	64 (22.1%)	0.238
Vaccination type			
Pfizer BioNTech	38 (60.3%)	186 (64.1%)	0.501
Oxford AstraZeneca	25 (39.6%)	100 (34.4%)	-
Unknown	0 (0)	4 (1.4%)	-

Categorical variables expressed as number (%) and *p*-value by Chi-square test. Continuous variables expressed as median (interquartile range) and *p*-value by Mann–Whitney U test.

**Table 4 vaccines-10-00498-t004:** Comparison of characteristics of breakthrough infections in vaccinated patients.

Total Patients *n* = 63 (Out of 353 Patients)	Before Omicron (1 April 2021 to 20 December 2021) *n* = 40	Omicron Period (21 December 2021 to 15 January 2022) *n* = 23	*p*-Value
Age, years	61 (52–74)	61 (40–74)	0.568
Caucasian	25 (62.5%)	14 (60.9%)	0.898
Gender, Male	23 (57.5%)	15 (65.2%)	0.547
Diabetes Mellitus	24 (60%)	10 (43.4%)	0.205
Hypertension	29 (72.5%)	19 (82.6%)	0.364
Cardiovascular disease	12 (30%)	7 (30.4%)	0.971
Cancer	3 (7.5%)	2 (8.7%)	0.866
Previous kidney transplant	2 (5%)	6 (26.1%)	0.016
Immunosuppression	8 (20%)	7 (30.4%)	0.349
Vaccination status			
Received first dose	40 (100%)	23 (100%)	-
Received second dose	40 (100%)	23 (100%)	-
Received third dose	4 (10%)	17 (74%)	-
Vaccination type			
Pfizer BioNTech	24 (60%)	14 (%)	-
Oxford AstraZeneca	16 (40%)	9 (%)	-
Antibody status in Nov 2021			
Positive	30	17	-
Not done	10	6	-
COVID-related hospitalisation	5 (12.5%)	0	-
COVID-related death	2 (5%)	0	-

## Data Availability

The data analysed in the current study are available from the corresponding author on reasonable request.

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
