# Peer review of "Humoral Response in Hemodialysis Patients Following COVID-19 Vaccination and Breakthrough Infections during Delta and Omicron Variant Predominance"

_vaccines, 2022, doi:10.3390/vaccines10040498_

Round 1

Reviewer 1 Report

This is a well-designed and written paper focused on the study on efficacy of   Astra-Zeneca and Pfizer mRNA-based vaccines against both Delta SARS-CoV-2 and Omicron variant. The paper has an outstanding importance in terms of understanding of the need of third vaccine dose administration. In view of my expertise I believe that the presented data may encourage people to receive a third dose of anti-COVID-19 vaccine. 

Author Response

Comments: This is a well-designed and written paper focused on the study on efficacy of AstraZeneca and Pfizer mRNA-based vaccines against both Delta SARS-CoV-2 and Omicron variant. The paper has an outstanding importance in terms of understanding of the need of third vaccine dose administration. In view of my expertise, I believe that the presented data may encourage people to receive a third dose of anti-COVID-19 vaccine. 

Answer: Thank you very much.

Reviewer 2 Report

The authors evaluate humoral immunity of hemodialysis patients and investigate of breakthrough infections after receiving two different COVID-19 vaccines. The major limitation of this report is the small numbers of studied patients, however, the authors already pointed out the necessary of caution in interpretation of results. The second of weak side of the paper is that the results were not well presented, which would be improved.

  1. In lines 42-43, "Of these 63 patients, five patients were admitted into hospital due to COVID-19 related illness and there were two reported deaths (delta period)." To be more accurate, suggest change to:

 Of these 40 patients in delta period, five were admitted into hospital and two reported deaths due to COVID-19 related illness.

  1. (Table 1) should be added at line 137 after the sentence "… the median age was 61 years (49-72). In line 151, the (Table 2) in fact should be Table 1. The (Table 2) should be added in line 162, after the sentence "…antibody status following vac-161 cination developed antibodies to SARS-CoV-2 by the end of July 2021". (Table 3) should be followed in line 166.
  2. According to table 6, before Dec 21, 2021, proxy Omicron, the total numbers of breakthrough cases were N 40, however, the cases numbers adding together in figure 2 is 72, how could this be?
  3. As the total number of patients is low, there is no difference of immune response affected by the two vaccines and the breakthrough infections seem not related to the co-morbidities of these patients, whether the authors may provide more detail information of the 5 covid-related hospitalisation and 2 death. It is not clear whether the poor outcomes of these cases are related to multiple co-morbidities of these patients. Specifically, whether did one or combination of several preexisting conditions affect the outcome of these cases. It would be better if authors can provide more detail information for each of these cases.

Author Response

Q1. In lines 42-43, "Of these 63 patients, five patients were admitted into hospital due to COVID-19 related illness and there were two reported deaths (delta period)." To be more accurate, suggest change to: Of these 40 patients in delta period, five were admitted into hospital and two reported deaths due to COVID-19 related illness.

Answer: Thank you. We have now updated the text as per the comment.

Q2. (Table 1) should be added at line 137 after the sentence "… the median age was 61 years (49-72). In line 151, the (Table 2) in fact should be Table 1. The (Table 2) should be added in line 162, after the sentence "…antibody status following vac-161 cination developed antibodies to SARS-CoV-2 by the end of July 2021". (Table 3) should be followed in line 166.

Answer: Thanks. We have now moved the Table 1 as recommended. Table 2 is now converted to a figure and Tables 3 & 4 merged as recommended by Reviewer 3.

Q3. According to table 6, before Dec 21, 2021, proxy Omicron, the total numbers of breakthrough cases were N 40, however, the cases numbers adding together in figure 2 is 72, how could this be?

Answer: Thank you for this comment.

The idea behind the Figure 2 was to demonstrate the time between the first and second dose vaccination to the outbreaks separately in two different colours. Previously we included only 36 breakthrough cases before the third dose vaccination in this figure (we excluded the 4 cases which occurred after the third dose vaccination).

To make this clear, we have now included all the 40 breakthrough cases in the figure and modified the footnote of the figure as below by deleting the phrase “third dose vaccination and”. It now appears as “Time in months between the first and second dose of COVID-19 vaccination and the forty COVID-19 breakthrough infections before the Omicron proxy period (December 21, 2021)”.

Figure 2 will appear now as Figure 3 now due to conversion of Table 2 to Figure 2

Q4. As the total number of patients is low, there is no difference of immune response affected by the two vaccines and the breakthrough infections seem not related to the co-morbidities of these patients, whether the authors may provide more detail information of the 5 covid-related hospitalisation and 2 death. It is not clear whether the poor outcomes of these cases are related to multiple co-morbidities of these patients. Specifically, whether did one or combination of several pre-existing conditions affect the outcome of these cases. It would be better if authors can provide more detail information for each of these cases.

Answer: Thanks. We acknowledge this point. As the number of events were low (5 hospitalisations and 2 deaths), there were no specific factors that can be highlighted in these cases.

Reviewer 3 Report

Dear authors, 

please find enclosed more detailed comments regarding your manuscript entitled "Humoral response in hemodialysis patients following COVID- 2 19 vaccination and breakthrough infections during Delta and 3 Omicron variant predominance. "

Despite the interest of a deep analyse of the response to vaccination in people at high risk of severe SARS CoV 2 infection, the manuscript has some major limitations and needs to be revised in order to strengthen the results. 

Best regards, 

The manuscript submitted by R. Chinnadurai et al is a prospective observational cohort study from the central and satellite hemodialysis centers of Salfort Royal Hospital, United Kingdom. Four hundred and eleven patients were included in the study starting from january 2021. The objective was to analyse the humoral response to vaccination during the year 2021, which was characterized by the transition from « delta » to « omicron » period at its end. The percentage of seropositive patients and the breakthrough infections are presented in the first months following two doses vacination. It is interesting to have a systematic survey of the duration of Ab response afeter vaccination, and the protective effect against successive variants in such a population, which is known to have a weaker response to vaccines. However, the study has some limitations that render the conclusion rather

Specific Comments

  • The abstract could be shortened, it contains two many details and makes it difficult to get the main results of the study
  • The principal objective of the study is not clearly defined in the introduction. Is it to measure the percentage of seropositive patients in the months following vaccination ? is it to measure the prevalence of breakthrough infections after vaccination ? what was the scheduled duration of follow-up ?

Methods

  • The follow up of the patients could be described in more details (time points for IgG testing post vaccination), and the flow chart would be improved if the follow up was included.
  • Regarding the patients included in the final analysis, it is not clear whether those with positive Ab before vaccination were included. If so, this sub-group should be clearly analysed in the results
  • Details regarding the serology assay are lacking : does the assay detect anti-N and anti-S antibodies, or only anti-S, and is it really an IgG assay, or a total Ab assay including IgM and IgG ? To date, two Siemens SARS CoV-2 immunoassays are commercialized : the COVT assay, which is a qualitative assay detecting both IgG and IgM ; and the sCOVG, which is a semi-quantitative one. If it is the later, a conversion factor between sCOVG index and the WHO 20/136 International Standard binding antibody units (BAU)/mL has been established as: (sCOVG index)*21.8 = 1 BAU/mL (see. This point could be clarified by the authors, and eventually the results completed with semi-quantitative data if available.

Results :

  • Tables are not easy to read. For instance, in table 2, there is no major interest to have the absolute number of positive patients monthly. A graphical representation instead of a such a table, with cumulative percentages of seropositive patients according to the delay from vaccination would be preferable.
  • Table 3 and 4 could be merged in a single one.
  • It is not easy to interpret the statistical analysis of seropositive versus seronegative patients, as the second group is very small (12). This point is pointed out by the authors in the discussion part
  • Patients with previous Ab should appear in the breakthrough infections part (table 4 and 5), as it is interesting to know if these patients, who had a natural infection + received two doses of vaccine, were at lower risk of infection during the study. Moreover, it would have provide data about the re-infection frequency (even if the number of patients is quite low).
  • The way figure 2 is presented is not very clear. The addition of the total number of positive cases is 72, when it should be 40 (positives cases during the « delta » period). The authors should modify this figure in order to make it more easy to understand.

Discussion

- the authors clearly point out some of the limitations of their study, including the lack of homogeneity of the population, with small subgroups, and the absence of quantitative data regarding the Ab response.

Author Response

Specific Comments

Q1. The abstract could be shortened, it contains two many details and makes it difficult to get the main results of the study.

Answer: Thanks. We have now removed lines 31 to 33 to shorten the abstract and make it clear.

Q2. The principal objective of the study is not clearly defined in the introduction. Is it to measure the percentage of seropositive patients in the months following vaccination? is it to measure the prevalence of breakthrough infections after vaccination? what was the scheduled duration of follow-up?

 Answer: Thanks. We have now modified the final paragraph of the introduction to make the two major aims of this project clear.

All patients were followed up until January 15, 2022 (We have included a point in line 119 of the methods section to make this point clear).

Methods

Q3. The follow up of the patients could be described in more details (time points for IgG testing post vaccination), and the flow chart would be improved if the follow up was included.

 Answer: As included in the assays sub-section of the method section of manuscript (lines 126 to 132), all patients had monthly COVID-19 antibody testing and weekly rRT-PCR throat and nasal swabs. We have now included this point as a box in the flowchart as recommended. We followed all patients until January 15, 2022 as responded to question 2.

Q4. Regarding the patients included in the final analysis, it is not clear whether those with positive Ab before vaccination were included. If so, this sub-group should be clearly analysed in the results.

Answer: We acknowledge this important point. Yes, patients with baseline positive antibody were included. The pre-vaccination antibody status was negative in a greater proportion of patients who had breakthrough infections. We have now included this in table 3 and in the results section (lines 218 to 220).

Q5. Details regarding the serology assay are lacking: does the assay detect anti-N and anti-S antibodies, or only anti-S, and is it really an IgG assay, or a total Ab assay including IgM and IgG? To date, two Siemens SARS CoV-2 immunoassays are commercialized: the COVT assay, which is a qualitative assay detecting both IgG and IgM; and the sCOVG, which is a semi-quantitative one. If it is the later, a conversion factor between sCOVG index and the WHO 20/136 International Standard binding antibody units (BAU)/mL has been established as: (sCOVG index)*21.8 = 1 BAU/mL (see. This point could be clarified by the authors, and eventually the results completed with semi-quantitative data if available.

 Answer: Thanks. The assay used was the COV2T, detecting total antibodies (IgG and IgM) to SARS-CoV-2. We have included the details of this assay in the assay subsection of the method section.

Results:

Q6. Tables are not easy to read. For instance, in table 2, there is no major interest to have the absolute number of positive patients monthly. A graphical representation instead of a such a table, with cumulative percentages of seropositive patients according to the delay from vaccination would be preferable.

 Answer: Thanks. We have now converted Table 2 to Figure 2 to make this clear.

Q7. Table 3 and 4 could be merged in a single one.

 Answer: Thanks. We have now merged Tables 3 and 4 and now represented as ‘Table 2: Analysis A and Analysis B’. 

Q8. It is not easy to interpret the statistical analysis of seropositive versus seronegative patients, as the second group is very small (12). This point is pointed out by the authors in the discussion part.

 Answer: Thanks. We acknowledge this comment.

Q9. Patients with previous Ab should appear in the breakthrough infections part (table 4 and 5), as it is interesting to know if these patients, who had a natural infection + received two doses of vaccine, were at lower risk of infection during the study. Moreover, it would have provide data about the re-infection frequency (even if the number of patients is quite low).

Answer: Many thanks. As responded to Question 4, we acknowledge this very important point. We have now analysed the double-vaccinated group based on the pre-vaccination antibody status and noted that the breakthrough infections were significantly lesser in the group who had positive antibody status prior vaccination. We have highlighted these results in the results section of our revised manuscript and have included a Supplementary Table 1 to note this point.

Q10. The way figure 2 is presented is not very clear. The addition of the total number of positive cases is 72, when it should be 40 (positives cases during the « delta » period). The authors should modify this figure in order to make it more easy to understand.

 Answer: Thank you for this comment. As responded to Q3 of reviewer 2:

The idea behind the figure 2 was to demonstrate the time between the first and second dose vaccination to the outbreaks separately in two different colours. Previously we included only 36 outbreak cases before the third dose vaccination in this figure (we excluded the 4 cases which occurred after the third dose vaccination).

To make this clear, we have now included all the 40 outbreaks cases in the figure and modified the footnote of the figure as below by deleting the phrase “third dose vaccination and”. It appears now as “Time in months between the first and second dose of COVID-19 vaccination and the forty COVID-19 breakthrough infections before the Omicron proxy period (December 21, 2021)”.

Discussion:

Q11. The authors clearly point out some of the limitations of their study, including the lack of homogeneity of the population, with small subgroups, and the absence of quantitative data regarding the Ab response.

Answer: Thank you for your comment.

Round 2

Reviewer 3 Report

Dear authors,

Thank you for the improvement of the manuscript. 

This study provides useful data on humoral status in the first months following SARS-CoV-2 vaccination in a cohort of hemodialysis patients and contribute to the epidemiological description of breakthrough infections after vaccination. 

I have a few minor comments left on this revised manuscript:

  • "assays": instead of simply adding a sentence with the name of the assay, the paragraph should have been re-written in order to fusion line 103 and line 105-106.
  • figure 2: the axis titer on figure A must be on the left; please add a more detailed legend, in order to explain that figure B correspond to the percentage of patients with available serology data, i.e 281 (and not all the patients from figure A)
  • line 181: to my opinion, "lack of antibody detection" should be used instead of "attenuated antibody response", as the assay only provide qualitative data, but not any quantitative or functional information about the humoral response.
  • table 2: using "analysis A" and "analysis B" as titles is not very informative. At least Analysis B should be changed to "logistic regression analysis"
  •  figure 3: if I understand correctly the figure, all the breakthrough infections occurred after the second dose. Is there really an interest to add the delay from first dose?

Author Response

Response to reviewer’s comments

Q1: "assays": instead of simply adding a sentence with the name of the assay, the paragraph should have been re-written in order to fusion line 103 and line 105-106.

Answer: Thanks. We have modified this sentence to make it clear.

“All HD patients included in this study had been receiving monthly COVID-19 immunoglobulin G (IgG) antibody surveillance testing with Siemens’ COV2T immunoassay which is used to detect total antibodies (IgG and IgM) to SARS-CoV-2. This is a qualitative assay with an index >/= 1.0 being considered as positive based on the manufacturer-determined cut-off”.

Q2: figure 2: the axis titer on figure A must be on the left; please add a more detailed legend, in order to explain that figure B correspond to the percentage of patients with available serology data, i.e 281 (and not all the patients from figure A)

Answer: Thanks. We have included these changes in the figure and the legend to make it clear.

Q3: line 181: to my opinion, "lack of antibody detection" should be used instead of "attenuated antibody response", as the assay only provide qualitative data, but not any quantitative or functional information about the humoral response.

Answer: Thanks. We have reworded line 181 as "lack of antibody response"  to make it clear.

Q4: table 2: using "analysis A" and "analysis B" as titles is not very informative. At least Analysis B should be changed to "logistic regression analysis"

Answer: Thanks. We have included analysis A as comparative analysis and analysis B as logistic regression analysis in table 2.

Q5: figure 3: if I understand correctly the figure, all the breakthrough infections occurred after the second dose. Is there really an interest to add the delay from first dose?

 Answer: Thanks. We have now removed the first dose details in figure 3.